# Coupling spatial segregation with synthetic circuits to control bacterial survival

Shuqiang Huang[†,1], Anna Jisu Lee[†,1], Ryan Tsoi[1], Feilun Wu[1], Ying Zhang[2], Kam W Leong[2] & Lingchong You[1,3,*]

## Abstract

Engineered bacteria have great potential for medical and environmental applications. Fulfilling this potential requires controllability over engineered behaviors and scalability of the engineered systems. Here, we present a platform technology, microbial swarmbot, which employs spatial arrangement to control the growth dynamics of engineered bacteria. As a proof of principle, we demonstrated a safeguard strategy to prevent unintended bacterial proliferation. In particular, we adopted several synthetic gene circuits to program collective survival in *Escherichia coli*: the engineered bacteria could only survive when present at sufficiently high population densities. When encapsulated by permeable membranes, these bacteria can sense the local environment and respond accordingly. The cells inside the microbial swarmbot capsules will survive due to their high densities. Those escaping from a capsule, however, will be killed due to a decrease in their densities. We demonstrate that this design concept is modular and readily generalizable. Our work lays the foundation for engineering integrated and programmable control of hybrid biological–material systems for diverse applications.

**Keywords** collective survival; engineered bacteria; safeguard control; spatial segregation

**Subject Categories** Synthetic Biology & Biotechnology; Quantitative Biology & Dynamical Systems

**Mol Syst Biol. (2016) 12: 859**

## Introduction

A major endeavor in synthetic biology is to program functions of cells and cell populations in a predictable manner (Leonard *et al*, 2008; Khalil & Collins, 2010; Ruder *et al*, 2011; Riccione *et al*, 2012; Choi & Lee, 2013; Church *et al*, 2014; Nielsen *et al*, 2014; Olson & Tabor, 2014; Purcell & Lu, 2014; Andries *et al*, 2015; Ford & Silver,

2015). This ability is critical for realizing the full potential of synthetic biology for diverse applications in medicine, environment, and biotechnology (Benner & Sismour, 2005; Leduc *et al*, 2007; Keasling, 2008; Ruder *et al*, 2011; Weber & Fussenegger, 2012). To date, efforts in synthetic biology have focused on engineering of circuit-centric or cell-centric functions (Endy, 2005; Brophy & Voigt, 2014; Pardee *et al*, 2014). When engineering a circuit, the host cell is typically considered a static "chassis", or a reactor for the biochemical reactions associated with the circuit function. However, this practice is in contrast with operation of natural systems, where functions of cells or cell populations often result from detection of and response to their spatial arrangement (Pai & You, 2009). Depending on environmental conditions, including stress, nutrient, and temperature, many bacteria can form spatial structures consisting of cell aggregates, such as biofilms (Claessen *et al*, 2014) and fruiting bodies (O'Toole *et al*, 2000; Claessen *et al*, 2014) that allow them to achieve functions beyond the capability of individual cells or homogenous cell populations. We reason that this property represents a novel design strategy to engineer sophisticated functions in engineered cell populations.

Drawing inspiration from natural systems, we sought to develop a platform technology—microbial swarmbot, to demonstrate a predefined system function by integrating synthetic biology, biomaterials engineering, and microfluidics. A microbial swarmbot consists of a small population of engineered bacteria encapsulated in a polymeric capsule. Each swarmbot capsule works as an active carrier of engineered bacteria by coupling of cell-centric functions and environmental sensing. In such a system, bacteria can reside in two different compartments (inside or outside the capsule), which can serve as cues to trigger different dynamics.

As a proof of principle, here we demonstrate a safeguard strategy to prevent unintended bacterial proliferation (Church, 2005; Serrano, 2007). In particular, we use synthetic gene circuits to program collective survival in *Escherichia coli*, such that the engineered bacteria only survive when present at a sufficiently high density. When residing in the swarmbot capsules, these bacteria will survive due to their high local density. If they escape from the swarmbots, however, they will be killed due to a decrease in their

1 Department of Biomedical Engineering, Duke University, Durham, NC, USA
2 Department of Biomedical Engineering, Columbia University, New York, NY, USA
3 Center for Genomic and Computational Biology, Duke University, Durham, NC, USA
*Corresponding author. Tel: +1 919 660 8408; E-mail: you@duke.edu
†These authors contributed equally to this work

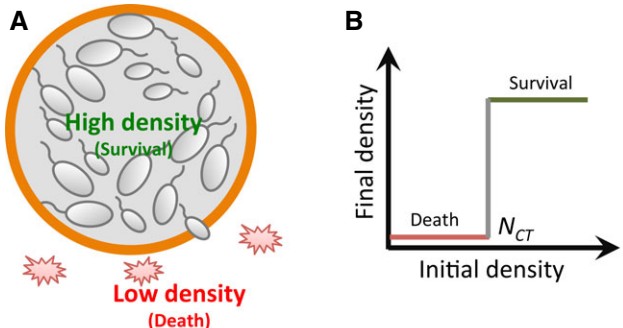

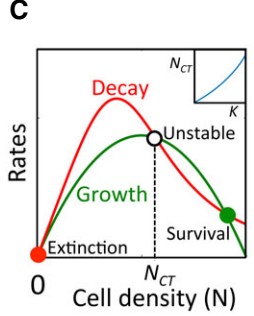

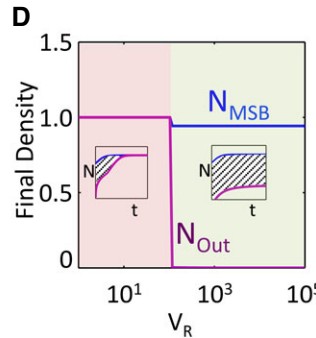

**Figure 1. Design and modeling of safeguard control in microbial swarmbots.**

A  Design concept. Bacteria are engineered to exhibit collective survival. Bacteria confined in the microbial swarmbot can maintain a high local density and survive. Cells escaping the swarmbot will have a reduced density due to a larger extra-capsule environment. If their density drops below their survival threshold, they will die, leading to safeguard control.

B  Collective survival. A population survives only when its initial density is > $N_{CT}$.

C  Simulation of collective survival. The collective survival can be described by a simplified model: $\frac{dN}{dt} = \mu N \left(1 - \frac{N}{N_m}\right) - \left(\frac{dK^\alpha}{N^\alpha + K^\alpha}\right)N$, where $\mu$ is the maximum growth rate of the population, $N_m$ is the carrying capacity of the population, $d$ is the maximum death rate of the cells, $K$ is the critical threshold for survival, and $\alpha$ is the Hill coefficient. We assume that the growth rate of cells follows the logistic equation and that the death rate decreases with $N$. As the initial conditions, the parameters are set as $N_O$ = variable from 0 to 1, $\mu = 1.75$, $N_m = 1$, $d = 2$, $K = 0.5$, and $\alpha = 4$. For the given parameter set, the system has three steady states. The trivial steady state (filled red circle) corresponds to extinction. The high state (filled green circle) corresponds to survival. Both are stable and are separated by an unstable steady state ($N_{CT}$, open circle). The population would reach the high steady state if and only if its density is greater than $N_{CT}$. The inset shows the correlation between $N_{CT}$ and model parameter $K$.

D  Modulating safeguard with $V_R$. The effectiveness of safeguard can be quantified by the area under the curve (AUC) from density differences between the microbial swarmbot and the surrounding environment. The swarmbot dynamics can be described by a two-compartment model:

$\frac{dN_{MSB}}{dt} = \mu N_{MSB} \left(1 - \frac{N_{MSB}}{N_m}\right) - \left(\frac{dK^\alpha}{N_{MSB}^\alpha + K^\alpha}\right) N_{MSB} - f_N(N_{MSB} - N_{Out})$,

and $\frac{dN_{Out}}{dt} = \mu N_{Out} \left(1 - \frac{N_{Out}}{N_m}\right) - \left(\frac{dK^\alpha}{N_{Out}^\alpha + K^\alpha}\right) N_{Out} + \frac{1}{V_R} f_N(N_{MSB} - N_{Out})$, where the transport of cells between the two compartments is described by the final term in each equation. Modeling predicts that AUC increases with $V_R$, and once $V_R$ becomes sufficiently large, AUC approaches infinity, indicating high level of safeguard. As the initial conditions, the parameters are set as follows: $N_{MSB} = 0.1$ at $t = 0$, $\mu = 1.75$, $N_m = 1$, $d = 2$, $K = 0.01$, $\alpha = 4$, $f_N = 0.1$, $V_R$ = variable from 1 to $10^5$. Insets demonstrate the simulated growth dynamics for two representative $V_R$ values.

densities (Fig 1A). We used several synthetic gene circuits to demonstrate the overall function, which underscores the modularity of the design concept. In our system design, we use the term *swarm* to emphasize the de-centralized nature of the *collective behavior* by the engineered bacteria. It does not imply a requirement for the cells to be highly motile.

# Results

### Design concept

Our central design concept in engineering safeguard control is to couple collective survival and environment sensing. Collective survival allows the population to survive only when its density is greater than a critical threshold ($N_{CT}$) (Fig 1B). For engineered bacteria, the population dynamics can be captured by an ordinary differential equation model. With appropriate parameters, the system can have three steady states (Fig 1C). The leftmost one is a trivial steady state, which corresponds to population extinction. The second ($N = N_{CT}$) is unstable. The rightmost corresponds to population survival. If the initial population density is below $N_{CT}$, the population will go extinct. Otherwise, it will survive and grow to a high final density. In subsequent simulations, the parameter $K$ is used to account for the difference in $N_{CT}$ of the population (Fig 1C, inset).

Environment sensing can be established by encapsulation of a population, which can sense the confinement (Fig 1A). We consider the interior of the microbial swarmbots as one compartment ($V_1$) and the exterior as another ($V_2$). Depending on the specific environment it resides in, the survival of a population can be determined by its local density. These dynamics can be described by a two-compartment model (Fig 1D). Our simulation predicts that the volume ratio between the two compartments ($V_R = V_2/V_1$) plays a critical role in safeguard performance, assuming that we start with a sufficiently high initial density in the swarmbots. At a small $V_R$, both cell densities inside the swarmbot ($N_{MSB}$) and outside ($N_{Out}$) can reach a high value, corresponding to no safeguard. At a sufficiently large $V_R$, $N_{MSB}$ reaches a high state but $N_{Out}$ cannot, leading to safeguard control (Fig 1D). $V_R$ can be modulated by adjusting $V_1$ or $V_2$.

### Programming collective survival

The safeguard control can be implemented by different systems that program collective survival. Experimentally, we program collective survival using three different gene circuits, which allow us to evaluate the modularity of the overall system design. To enable flexible environmental control of the programmed safeguard, all these circuits rely on the interplay between an engineered survival gene circuit and the presence of an antibiotic. In general, we note that the use of an antibiotic is not a critical requirement, but rather an implementation strategy to demonstrate the general design concept.

The BlaM circuit (Fig 2A) consists of constitutive expression of a modified $\beta$-lactamase (BlaM). Due to the deletion of periplasmic localization sequence that exists in the wild-type Bla (Tanouchi *et al*, 2012), BlaM can degrade a $\beta$-lactam antibiotic, such as carbenicillin (Cb), only when it is released from the lysed cells

(Appendix Fig S1A). If the initial cell density is sufficiently high, sufficient BlaM will be released to degrade the Cb to a sub-lethal concentration, allowing the remaining cells to survive.

The QS-CAT circuit (Fig 2B) programs expression of a chloramphenicol acetyltransferase (CAT) (Collins *et al*, 2006) under control of quorum sensing (QS). QS is a mechanism utilized by bacteria to sense and respond to changes in their densities (Fuqua *et al*, 1994; Singh *et al*, 2000; Waters & Bassler, 2005; Danino *et al*, 2010; Wu *et al*, 2014). It has been widely used to program dynamics in one (You *et al*, 2004; Pai *et al*, 2012; Smith *et al*, 2014) or multiple bacterial populations (Brenner *et al*, 2007; Balagadde *et al*, 2008; Song *et al*, 2009). Expression of CAT at a high cell density can

enable cells to survive in the presence of chloramphenicol (Appendix Fig S1B). A low-density population will go extinct due to lack of CAT expression.

The QS-BlaM circuit (Pai *et al*, 2012) (Fig 2C) consists of QS-mediated expression of BlaM fused to a sequence of HlyAs secretion tag, which is secreted by the HlyB/D transporters to mediate transport of the BlaM from the cytoplasm to the extracellular environment (Chervaux *et al*, 1995; Pai *et al*, 2012; Tanouchi *et al*, 2012). BlaM can also be released due to lysis of some cells, at an initially lethal antibiotic concentration. Notably, QS-BlaM circuit requires an extra step—secretion of BlaM—that can delay the utilization of BlaM (Appendix Fig S1C).

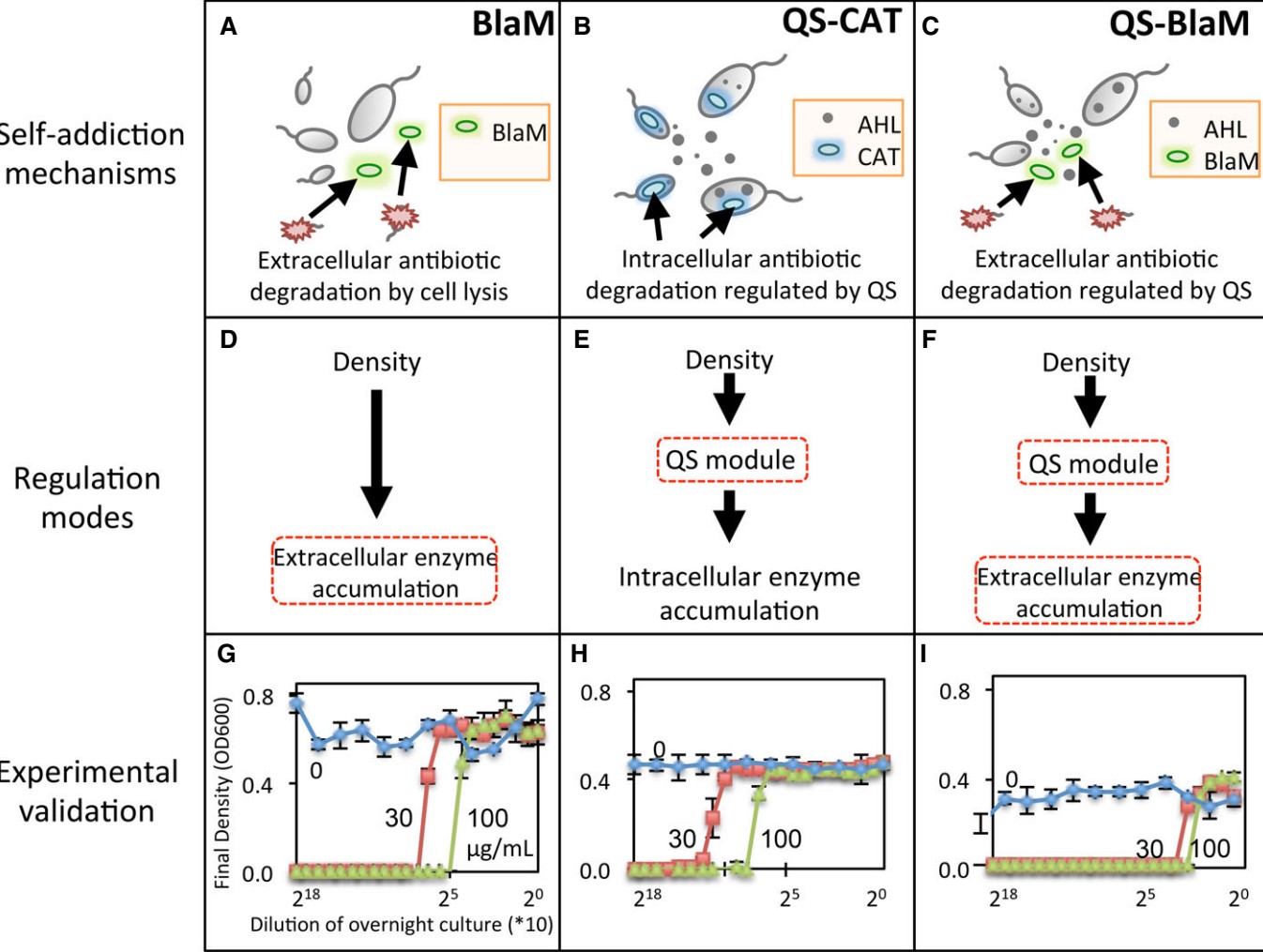

**Figure 2. Programming collective survival using different circuits.**

A–C   Mechanisms of three collective survival circuits. The BlaM circuit consists of constitutive expression of BlaM to degrade carbenicillin upon cell lysis. BlaM works as an extracellular enzyme. The QS-CAT consists of QS-mediated expression of a chloramphenicol acetyltransferase (CAT) that degrades chloramphenicol. The QS-BlaM circuit integrates QS control and extracellular carbenicillin degradation to establish collective survival.

D–F   Regulation modes for the three circuits. The BlaM circuit depends on antibiotic-mediated release of an enzyme into the environment to realize the collective survival. The QS-CAT circuit depends on QS regulation. The QS-BlaM circuit depends on both QS regulation and enzyme release (by lysis and export).

G–I   Experimental validation of collective survival. All populations grew to high cell densities in the absence of antibiotics (blue curves). In the presence of antibiotics, only populations with initial densities greater than $N_{CT}$ grew. $N_{CT}$ was increased with the concentration of antibiotics. Engineered bacteria with the QS-BlaM circuit exhibited a significantly higher $N_{CT}$ than those with the BlaM or the QS-CAT circuits. Carbenicillin was used for the BlaM and QS-BlaM circuits, and chloramphenicol, for the QS-CAT circuit. Each error bar represents the standard deviation from triplicate measurements.

Due to the differences in their implementation, we expect the different circuits to exhibit different survival threshold ($N_{CT}$) for collective survival. For the BlaM circuit, BlaM has to be released into the *extracellular* space to degrade the antibiotic (Fig 2D). In the QS-CAT circuit, CAT can be utilized *intracellularly* under the control of QS mechanism (Fig 2E). In the QS-BlaM circuit (Fig 2F), BlaM is expressed only at a sufficiently high initial cell density by the QS regulation. Even when BlaM is fully expressed in each cell, enough cells have to be present to produce and secrete sufficient BlaM to enable population survival.

Our experiments confirmed collective survival by engineered bacteria carrying each circuit (Fig 2G–I). In the presence of carbenicillin, cells carrying the BlaM circuit grew to high final densities only if their initial densities were sufficiently high (< 20,480-fold dilution of calibrated overnight culture for 30 μg/ml Cb, while < 2,560-fold for 100 μg/ml Cb). Those with lower initial densities did not grow. The control strain without the BlaM circuit did not grow regardless of the initial densities (Appendix Fig S2A). In the absence of Cb, all cultures grew to high final densities, regardless of their initial densities. For each circuit, the survival threshold ($N_{CT}$) increased with the concentration of the appropriate antibiotic. Engineered bacteria carrying the QS-CAT or QS-BlaM circuits exhibited similar behaviors as the BlaM circuit. However, cells carrying the QS-BlaM circuit exhibited a higher survival threshold than cells carrying the other two circuits (< 320-fold dilution for 30 μg/ml Cb and < 160-fold for 100 μg/ml Cb). The higher $N_{CT}$ by QS-BlaM cells could be due to several factors. In comparison with constitutive expression of BlaM in cells carrying the BlaM circuit, cells carrying the QS-BlaM circuit rely on QS to activate expression of BlaM. Indeed, induction of QS led to significantly reduced $N_{CT}$ in the QS-BlaM circuit in comparison with the uninduced condition (Appendix Fig S13). In addition to QS control, the inefficient export of BlaM by HlyB/D apparatus or reduced enzymatic activity of BlaM through transportation could further increase the $N_{CT}$.

## Programming collective survival in the microbial swarmbot platform

We next implemented spatial arrangement of populations by encapsulating the engineered populations into hollow hydrogel microcapsules (Appendix Figs S3 and S4), as basic units for microbial swarmbots. Bacteria could escape through intrinsic imperfections in the capsule membrane (Klinkenberg *et al*, 2001). These characteristics of encapsulation allowed us to test the effectiveness of engineered safeguard control that concerns whether escaping cells could survive.

To investigate the system dynamics in two compartments segregated by encapsulation (APA membrane), we developed a full model, specific for the BlaM circuit to account for the essential reactions involved in the circuit dynamics (equations 3.1–4.6, Code EV1). These include cell growth, lysis by Cb, release of BlaM, and degradation of Cb by BlaM. Also, we account for the transport of nutrient, Cb, BlaM proteins, and cells across the shell of swarmbot capsule (Fig 3A). The simulations using the model with varying initial densities conform to the collective survival of engineered bacteria (Appendix Fig S5) that were demonstrated by liquid culture experiments (Fig 2G): at a sufficiently high initial antibiotic concentration ($a = 0.2$), the encapsulated bacteria can only survive when their initial density in the swarmbot is greater than a threshold ($N_{CT}$,

Appendix Fig S5), which increases with $a$. Starting from a high initial density, the bacteria in the swarmbot will initiate growth first (Fig 3B, solid green line): this high-density population survives by establishing a higher local BlaM concentration to reduce the local Cb concentration (Appendix Fig S6). Growth of escaped cells will be initially suppressed due to a much higher Cb-mediated killing activity in the chamber (Fig 3B, dashed green line). When starting at a low initial density, bacteria cannot survive in the swarmbot or in the chamber (Fig 3C, red lines). In contrast, bacteria can grow both in the swarmbot and in the chamber in the absence of the antibiotic, regardless of their initial cell densities (Fig 3C).

To examine our predictions, we designed and fabricated a microfluidic device that enables dynamic control and monitoring of population dynamics of engineered bacteria in the swarmbots (Fig 3D). The swarmbot capsules were loaded into the culturing chamber with appropriate media and cultured at 37°C. When starting from a sufficiently high initial density in the capsule, bacteria carrying the BlaM circuit survived and grew in the swarmbots and, after a delay, in the chamber when treated with 100 μg/ml Cb (Fig 3G). GFP fluorescence was quantified from microscopy to demonstrate the dynamics over time (Fig 3E and Appendix Fig S9A, green lines). Under the same condition, a microbial swarmbot containing the engineered bacteria at a low initial cell density could not initiate growth either in the swarmbot or in the chamber (Fig 3E and Appendix Fig S9A, red lines), indicating collective survival. In the absence of the antibiotic, swarmbots with high or low initial densities initiated growth both in the swarmbots and in the chambers (Fig 3F and Appendix Fig S9B). In contrast, even starting from a high initial density, bacteria lacking the BlaM circuit did not grow in the swarmbot when treated with 100 μg/ml Cb (Fig 3H and Appendix Fig S2B). That is, the collective survival was indeed due to the BlaM circuit.

Both modeling and experiment demonstrated the presence of transient safeguard for a microbial swarmbot with a high initial cell density when treated with a sufficiently high antibiotic concentration. The presence of antibiotic caused a more prolonged delay in the growth in the chamber, in comparison with the delay in the absence of antibiotic (Fig 3). However, this safeguard performance is limited due to the eventual and significant growth in the chamber. The transient nature of the safeguard makes intuitive sense. Due to the lack of replenishment, the Cb in the system would eventually be degraded by BlaM released from lysed cells, as long as sufficient cells could survive in the swarmbots. As the Cb concentration in the chamber decreases to a sufficiently low level below the threshold ($a < 0.2$, Appendix Fig S6), the escaped bacteria can initiate growth and destroy the safeguard. From the perspective of the general design concept (Fig 1), a reduced antibiotic concentration corresponds to a reduced $N_{CT}$, making it more difficult to maintain safeguard. Another factor contributing to the quick loss of safeguard is the relatively small $V_R$ (the volume of the chamber to that of the capsule). The safeguard performance was further diminished due to decreased $V_R$ if multiple capsules were cultured together (Appendix Fig S7).

## Programming sustained safeguard using a pulsing condition

Both limitations can be overcome by adopting periodic pulsing of a fresh medium (Fig 4A, left), which simultaneously increases the

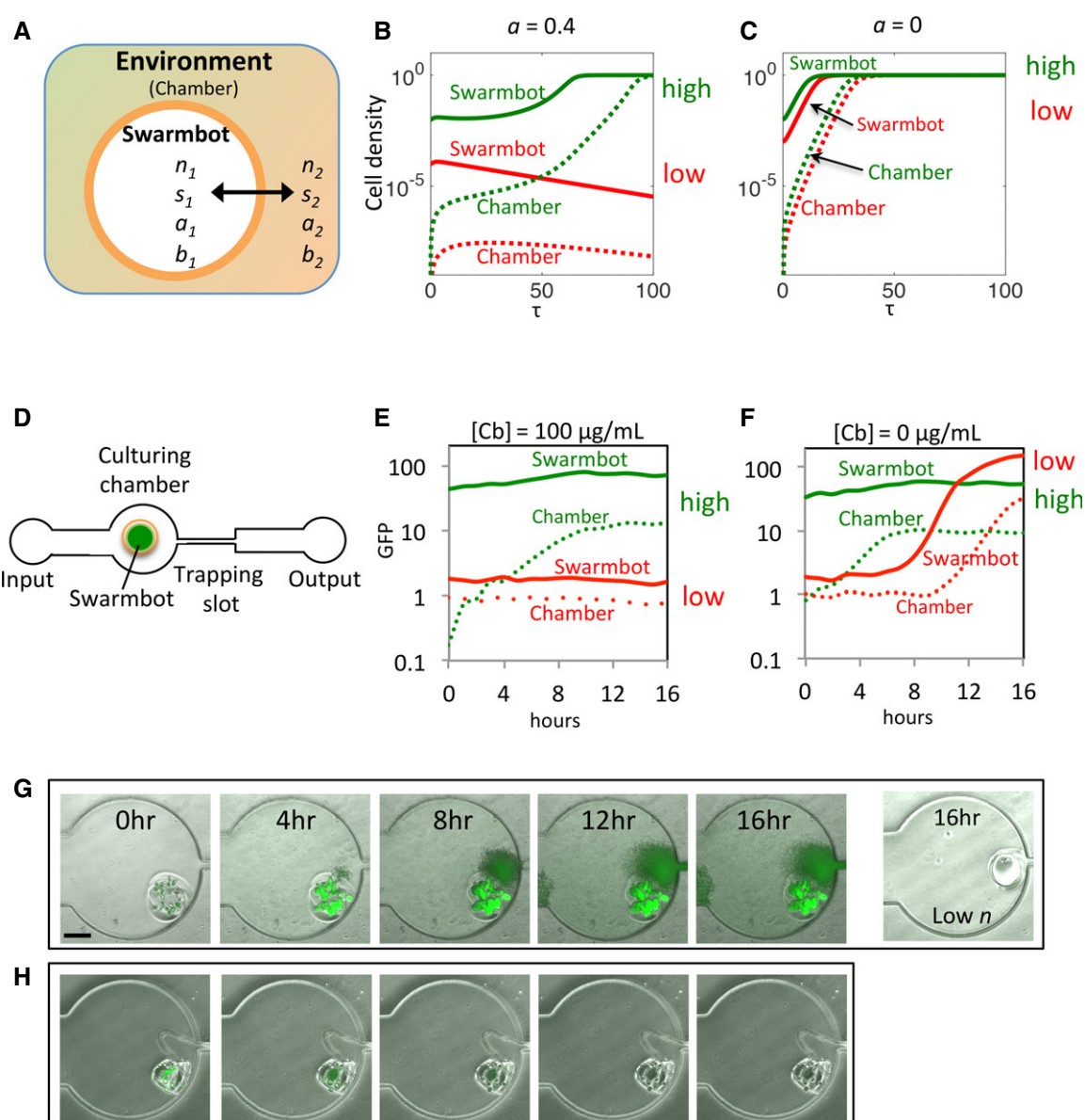

**Figure 3. Safeguard control in microbial swarmbot using the BlaM circuit.**

A       Configuration of the two-compartment model. The microenvironment of microbial swarmbot is separated from that of the chamber by its permeable hydrogel shell (orange), resulting in two separate compartments, defined as swarmbot and environment (chamber). Four major components involved in the population dynamics are explored in the model, including cell density ($n$), nutrient ($s$), carbenicillin ($a$), and BlaM concentration ($b$). Each component has a different transport rate, depending on the molecule size.

B,C     Simulated microbial swarmbot dynamics under a static condition. $a = 0.4$, a microbial swarmbot with a low $n_0$ cannot initiate growth (red lines) but one with a high $n_0$ leads to growth both in the swarmbot and, after a delay, in the chamber (green lines). (C) Swarmbots with both low and high initial densities lead to eventual full growth in the chamber in the absence of the antibiotic ($a = 0$). Solid lines: cell density inside the swarmbots; dashed lines: cell density in the chambers.

D       A microfluidic device for controlling and monitoring population dynamics of microbial swarmbot. The device consists of an input channel leading to a culturing chamber and a trapping slot. The chamber is used for culturing engineered bacterial populations both inside and outside the swarmbot, while the trapping slot is for trapping the swarmbot due to its smaller size but allowing the medium to flow through. Input flow patterns (static or pulsing) are controlled by a programmable syringe pump.

E,F     Experimental validation of model prediction. At 100 µg/ml carbenicillin, a microbial swarmbot with a high initial density led to robust growth in the capsule and, after a delay, in the chamber (green lines). While a swarmbot with a low initial cell density did not exhibit growth either in the swarmbot or in the chamber at the same carbenicillin concentration (red lines). In contrast, both the swarmbot and culturing chamber reached high cell densities in the absence of carbenicillin (F), regardless of the initial cell densities.

G       Time-lapse images of microbial swarmbot dynamics. The population initiated growth inside the swarmbot and subsequently in the chamber at a high initial density. But no growth occurred for a swarmbot with a low initial density at the same antibiotic concentration (rightmost). The concentration of carbenicillin was 100 µg/ml, and the scale bar represents 200 µm.

H       The control strain lacking BlaM circuit. The population could not survive at the same concentration of carbenicillin at the high initial density.

                                                                        

effective size of $V_2$ and maintains the $N_{CT}$. In our model, the pulsing flow is described with dilution and replenishment of each main component every time the culturing chamber is infused with a fresh medium (equations 3.1–4.6).

We modeled the effects of two experimentally controllable parameters on the safeguard performance under a pulsing condition: the antibiotic concentration ($I_a$) and the nutrient level ($I_s$) in the input flow. The antibiotic concentration can adjust the rate of cell killing and BlaM release that are critical for realizing the safeguard. The nutrient level modulates cell growth, which also indirectly affects the lysis rate as the faster growing cells are more susceptible

to Cb killing because Cb disrupts cell wall synthesis (Tuomanen *et al*, 1986). We use the area under the curve (AUC) of the difference between the final densities in swarmbots and in the chamber to quantify the degree of safeguard (Fig 4A, right): a larger AUC indicates more effective safeguard.

We simulated the dependence of AUC on combinations of antibiotic concentration ($I_a$) and nutrient level ($I_s$) supplemented in the fresh medium (Fig 4B). The resulting heatmap reveals three distinct states of safeguard performance. When $I_a$ is too small (no safeguard), bacteria can grow both in the swarmbot and in the chamber, causing loss of safeguard (also see Fig 4C, panels 1-3, for sample time courses). When $I_a$ is too large (Extinction), bacteria cannot grow in the swarmbot or in the chamber (also see Fig 4C, panels 8–9). However, for intermediate $I_a$ values (safeguard), bacteria grow in the swarmbot but not in the chamber, leading to robust safeguard (also see Fig 4C, panels 4–6). Also, $I_s$ plays a critical role in modulating the safeguard performance: the safeguard region shrinks with increasing $I_s$ (Fig 4B). This is in major part due to a reduced lysis rate when cells grow slower, thus allowing them to survive a higher antibiotic concentration (Fig 4C, panel 7).

Experimentally, we employed periodic dosing of fresh media with or without antibiotics to test the model predictions (see Materials and Methods). The experimental results were consistent with the model predictions and confirmed the parametric space to maximize safeguard performance (Fig 5A). An intermediate antibiotic concentration ([Cb] = 30 μg/ml) along with a lower nutrient concentration ([glucose] < 0.4%) was the most effective in maintaining the safeguard, as predicted by our model. Sample time courses show sustained GFP fluorescence in the swarmbots but negligible GFP in the chamber (Fig 5A). In addition, a high Cb concentration (100 μg/ml) or a high nutrient level ([glucose] = 0.4%) failed to maintain the population survival (Fig 5B and Appendix Fig S10). We note that the lack of growth in the chamber at a high antibiotic concentration was indeed due to circuit dynamics (Fig 5B, panels 4–9), instead of washout by the flow. This was evident in the growth of bacteria in the chamber, in the absence of the antibiotic (Fig 5B, panels 1–3). We would expect no growth in the chamber if washout had occurred (Materials and Methods).

## Modular control of safeguard

As proposed in Fig 2, different collective survival systems can be used to program safeguard. Indeed, we further demonstrated that engineered bacteria containing the QS-CAT and QS-BlaM circuits also achieved varying degrees of safeguard performance. Similar to the BlaM circuit, we established the safeguard control for QS-CAT bacteria by applying the same pulsing flow with 100 μg/ml chloramphenicol. Robust growth occurred in the swarmbot but not in the chamber. Cells carrying the QS-BlaM circuit exhibited safeguard control even under a static condition (Fig 6, Appendix Fig S11 and Movies EV1 and EV2). Over time, the escaping cells were eventually killed. This increased safeguard performance could be due to the higher $N_{CT}$ of QS-BlaM circuit compared with the other two systems that makes the collective survival more sensitive to the initial density (Fig 2G–I). The patchy growth observed in the absence of antibiotics (Fig 6 and Appendix Fig S12) was likely due to the spatial heterogeneity of the PDMS surface.

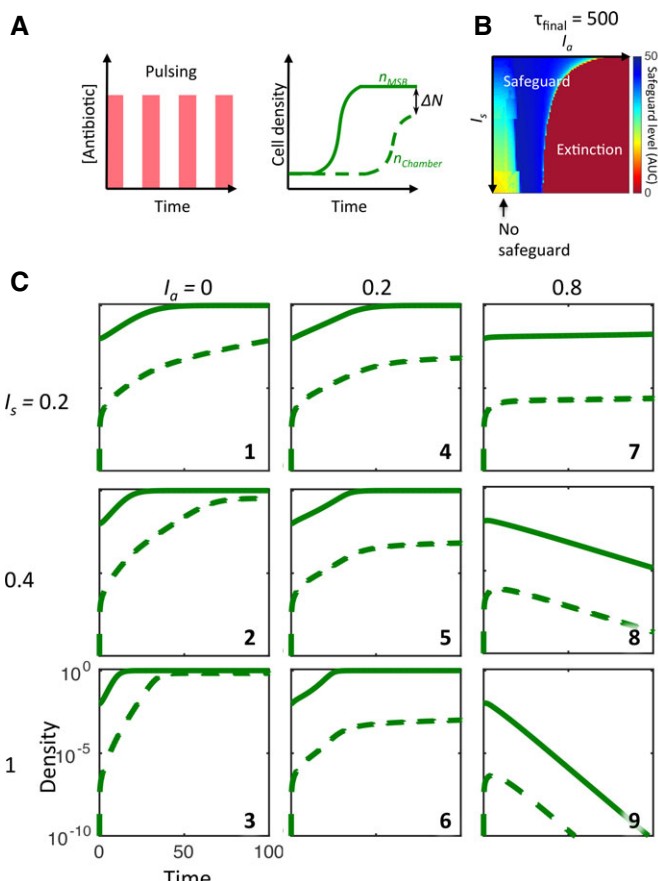

**Figure 4.  Modeling BlaM circuit-mediated microbial swarmbot safeguard under a pulsing condition.**

A   Left: a periodic pulsing condition. Right: quantification of safeguard effectiveness by evaluating the AUC of the difference between the cell density in the swarmbot and that in the chamber ($\Delta N$). A larger AUC corresponds to more effective safeguard.

B   Effectiveness of safeguard (AUC) as a function of the antibiotic concentration ($I_a$) and the nutrient concentration ($I_s$). The heatmap is divided into three regions: no safeguard—cells grow both in the swarmbot and in the chamber; safeguard—cells grow in the swarmbot but not in the chamber; extinction—cells do not grow in the swarmbot or in the chamber. For each $I_s$, safeguard is achieved for intermediate $I_a$ values. The range of $I_a$ values able to support safeguard is wider for smaller $I_s$ values.

C   Sample temporal swarmbot dynamics corresponding to different combinations of $I_a$ and $I_s$ values. Panels 1–3 correspond to no safeguard; panels 4–7 correspond to safeguard; panels 8-9 correspond to extinction.

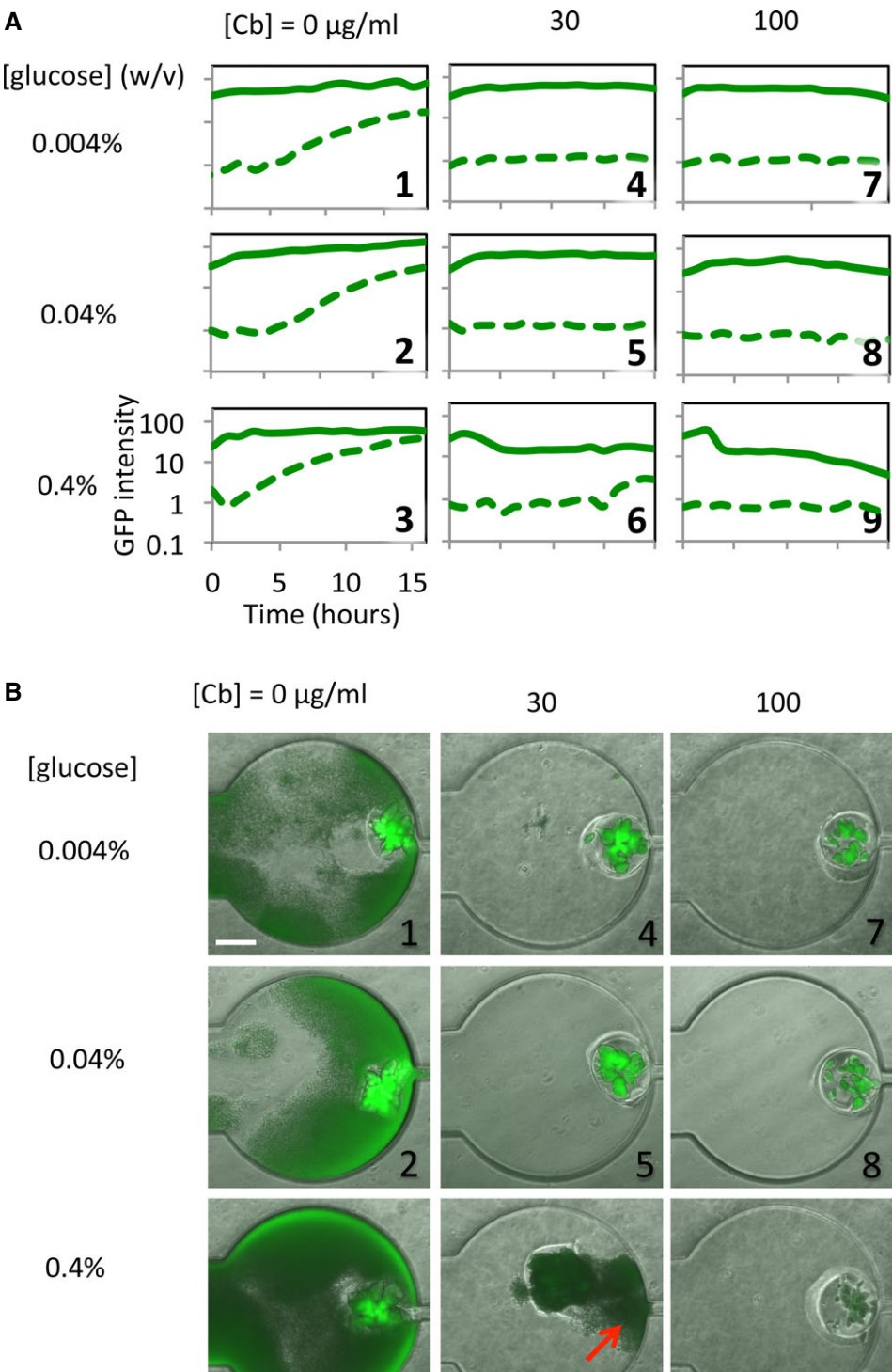

**Figure 5.  Experimental modulation of BlaM circuit-mediated safeguard control by controlling nutrient and antibiotic concentrations, under a pulsing condition.**

A   Time courses of swarmbots treated with varying concentrations of carbenicillin and glucose. Solid lines indicate cell densities inside the swarmbots; dotted lines indicate those in the chamber. Safeguard was established for lower nutrient levels (0.004% or 0.04% glucose) along with carbenicillin treatment. At a high nutrient level (0.4% glucose), the safeguard was lost due to either growth in the chamber (panels 3 and 6) or gradual killing of the population inside the swarmbot (solid line in panel 9). These results are qualitatively consistent with the model predictions (Fig 4).

B   Snapshots of safeguard performance at the end time point of (A). The red arrow in panel 6 indicates significant growth in the chamber; the dimmer fluorescence of the swarmbot in panel 9 indicates killing of the population.

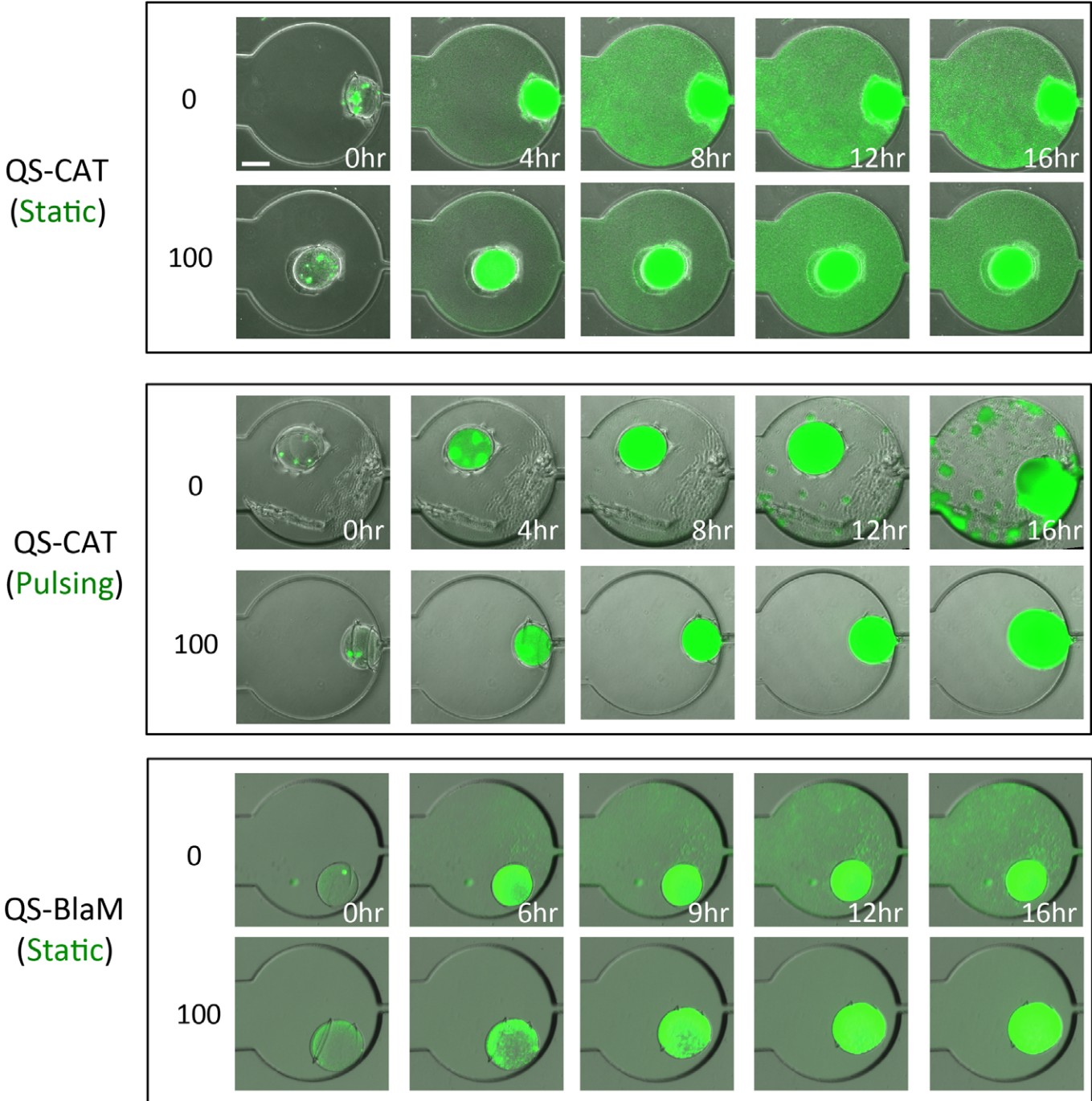

**Figure 6.  Experimental control of microbial swarmbot safeguard by other circuits.**
Demonstration of modular safeguard performance of swarmbots with different collective survival systems. For the swarmbots containing cells carrying the QS-CAT circuit, pulsing flow of medium containing 100 μg/ml chloramphenicol established safeguard control but not for the static condition. For the swarmbots with the QS-BlaM circuit, the static condition of 100 μg/ml carbenicillin was sufficient for safeguard control. For all the conditions, safeguard did not occur in the absence of antibiotic (top panels from each condition). All the images were from the swarmbots cultured at 37°C for 16 h. The scale bar is 250 μm.

## Discussion

These observed properties demonstrate that our microbial swarmbot technology embraces a new design strategy in programming sophisticated system functions. The microbial swarmbot implements an essential property of multicellular bacterial structures in nature (de la Fuente-Nunez *et al*, 2013)—a combination of spatial arrangement and dynamic environment sensing. While microencapsulation of cells has been adopted for diverse applications (Cruise *et al*, 1999; Chang, 2005), in most cases the capsules are passive carriers of the

encapsulated cells or enzymes. That is, the function of the encapsulated cells or enzymes is largely independent of the physical confinement by the capsules, other than the fact that encapsulation can modulate the lifetime or the release and transport rate of the encapsulated contents. These strategies heavily depend on passive diffusion or degradation of the encapsulation membrane for controlling the release of a drug or protein (Grayson *et al*, 2003).

In contrast, our approach involves coupling of encapsulation with the engineered cells as an integral part to achieve the overall system function: there is an active feedback between the circuit function and sensing of the microenvironment. Encapsulation creates a unique microenvironment different from the extra-capsular environment. This difference then triggers different bacterial population dynamics, leading to the designed safeguard control. The properties of the capsules and the culturing environment, including the permeability of the encapsulating material, the relative dimensions of the swarmbot capsule with respect to the culturing environment, and the dosing pattern of antibiotics, are all critical to realize the overall safeguard control, as are the parameters associated with the gene circuit carried by the engineered bacteria.

Another salient feature of our microbial swarmbot platform is its modularity, as demonstrated by qualitatively the same safeguard function realized by different gene circuits. Given this modularity, different aspects of the platform can be individually tuned, depending on objectives of the overall system. In this study, we have used antibiotics to enable flexible control of the extent of safeguard in different circuits (Fig 2), to demonstrate the fundamental design concept. However, the use of antibiotic is not in itself critical. Instead, any circuits that can realize collective survival in one or multiple populations can be embedded in our design framework. These include other engineered bacteria (Smith *et al*, 2014) or yeast cells (Dai *et al*, 2012) that only survive at sufficiently high initial cell densities or microbial populations exhibiting mutualistic interactions (Shou *et al*, 2007; Kerner *et al*, 2012). In these systems, the use of exogenous antibiotics is not a critical component of the engineered circuit dynamics. As such, our system can serve as a foundation for programming more complex dynamics resulting from "smart" hybrid biological-material systems.

In addition to the use of different circuits, diverse polymeric materials can be adopted to tune parameters critical for the overall system function, including permeability (Chan & Neufeld, 2010), mechanical properties (Kamata *et al*, 2014), and environmental responsiveness (Xia *et al*, 2013). The modular nature of our design can accommodate further optimization and extension by equipping the bacteria with effector genes of interest. These include effectors to target tumor cells (Anderson *et al*, 2006; Zhang *et al*, 2010), to kill bacterial pathogens (Saeidi *et al*, 2011; Gupta *et al*, 2013; Hwang *et al*, 2014), or to degrade environmental pollutants (Furukawa, 2000). For such applications, the programmed safeguard provides an intrinsic mechanism to prevent unintended proliferation of such engineered bacteria, a major safety concern that limits their broad applications.

# Materials and Methods

### Bacterial strains, growth media, and culturing conditions

*Escherichia coli* strains MC4100, TOP10, and MG1655 were used for carrying BlaM (Tanouchi *et al*, 2012), QS-CAT, and QS-BlaM

(Pai *et al*, 2012) circuits, respectively. All these strains were engineered with a GFP reporter. *E. coli* MC4100 lacking a collective survival circuit but with the GFP reporter was considered the control strain for the microbial swarmbot (Appendix Fig S2). Unless otherwise indicated, experiments were conducted in modified M9 medium [1× M9 salts (48 mM $Na_2HPO_4$, 22 mM $KH_2PO_4$, 862 mM NaCl, 19 mM $NH_4Cl$), 0.4% glucose, 0.2% casamino acids (Teknova), 0.5% thiamine (Sigma), 2 mM $MgSO_4$, 0.1 mM $CaCl_2$]. For overnight cultures, we inoculated single colonies from an agar plate into 4 ml Luria–Bertani (LB) medium (Genesee Scientific). Culture medium contained corresponding antibiotics (50 μg/ml chloramphenicol, 100 μg/ml Cb, and/or 100 μg/ml kanamycin) for plasmid maintenance. 0.5 mM IPTG, 0.1% arabinose, and 100 ng/ml aTc were used for induction of gene expressions in QS-BlaM circuit.

### Evaluation of collective survival for engineered bacteria

Engineered bacteria with BlaM, QS-CAT, and QS-BlaM circuits were inoculated in 4 ml LB containing appropriate antibiotics and cultured at 37°C with a shaking speed at 250 rpm for overnight. We calibrated initial densities of overnight cultures (OD600 of 0.5) by using a Victor 3 plate reader (Perkin Elmer). Then, we diluted the calibrated cultures into M9 medium ($10 \times 2^0$ to $10 \times 2^{22}$-fold), which was supplemented with varying concentrations of antibiotics (0, 30, and 100 μg/ml). About 200 μl cultures were aliquoted into 96-well plates, which were then sealed with two membranes, Aera-Seal (Excel Scientific) and Breathe-Easy membrane (Diversified Biotech), to prevent evaporation. The plates were fixed in a shaker at 250 rpm at 37°C for 24 h before cell-density measurements. For data analysis, the measured OD600 values were background corrected.

### Production of hollow alginate-poly (L-lysine)-alginate microcapsules

Alginate-poly (L-lysine)-alginate (APA) microcapsules containing bacterial cells were fabricated by a soft lithography technique (Duffy *et al*, 1998). The photoresist master mold was fabricated by spin-coating two layers of SU8 3035 (MicroChem, 1,000 rpm for 30 s) onto a silicon wafer, which was then pre-baked for 60 min at 95°C, and UV cross-linked with pre-designed photomask (AutoCAD). The UV exposed wafer was post-baked for 20 min at 95°C, developed with SU8 developer, and finally hard-baked for 2 h at 180°C. We then fabricated the PDMS template by pouring mixed PDMS solution (Dow Corning SYLGARD® 184, 10:1 as monomer: curing agent) onto the master mold, degassing under vacuum, and baking at 80°C for 30 min.

Before the fabrication of swarmbot capsules, optical density (OD600) of overnight culture was adjusted to 0.5 and diluted 1,000-fold into 1% alginate solution. The microcapsules were then fabricated following the layer-by-layer technique (De Koker *et al*, 2011; Delcea *et al*, 2011). The PDMS template was degassed in vacuum for 15 min and immersed in 1% alginate mixture containing different engineered bacteria. First, to polymerize alginate mixture in the wells, the PDMS template was transferred to a solution of 500 mM $CaCl_2$ (50 mM for bacteria with QS-BlaM circuit due to their higher susceptibility to $Ca^{2+}$ from our experimental observation) with 150 mM NaCl for 15 min. The polymerized alginate

microparticles were detached from the template with ultrasonic treatment for 1 min and coated with 0.05% poly (L-lysine) (PLL) solution on a tabletop shaker at 120 rpm for 10 min. To reduce reactivity of the positively charged PLL layer, another layer of 0.05% alginate solution was coated over the microparticles at 120 rpm for another 10 min. The cores of the microparticles were liquefied by chelating the $Ca^{2+}$ with a 100 mM sodium citrate solution at 120 rpm for 20 min. The microcapsules were washed twice between each coating with 0.85% NaCl to remove residual polymers. Each cylindrical microcapsule is ~250 μm in diameter, ~125 μm in height, and ~15 μm in shell thickness (Appendix Fig S4). Immediately after encapsulation, the swarmbot capsules were divided into two groups. One group was stored at 4°C (swarmbots with low initial densities) while the other group was pre-grown at 37°C for about 8 h for BlaM and QS-CAT and 10 h for QS-BlaM (swarmbots with high initial densities). The reason why we cultured longer time for bacteria with QS-BlaM was they grew slower than those with the other two circuits. This pre-growth stage allows the cells inside the swarmbots of the second group to reach a high density, which was estimated by the population size under microscopy. Unless otherwise noted, we use the swarmbot capsules from the second group for experiments with microfluidic device.

### Fabrication of microfluidic devices

We used a similar soft lithography technique to fabricate the microfluidic device. Three layers of SU3035 were spun onto a silicon wafer to fabricate the mold. The height of the SU8 mold for microfluidic device was ~450 μm. We then fabricated the PDMS microfluidic device containing the designed structure from the silicon mold and drilled holes for input and output flow with biopsy punches (World Precision Instruments Inc.). The PDMS device was bonded to a cover slip by using plasma treatment.

The main structure of the microfluidic device consists of input reservoir (diameter = 2 mm), main channel (width = 500 μm, length = 5 mm), culturing chamber (diameter = 1 mm), trapping channel (width = 50 μm, length = 1 mm), and output reservoir (diameter = 0.75 mm). The height of PDMS slab is about 4 mm; thus, the total volume of microfluidic device ($V_m$) is calculated as about 20 μl. The swarmbots are captured in the culturing chamber since the dimension of microcapsules is larger than the trapping channel, which prevents them from flowing out of the microfluidic device.

### Culturing and monitoring of microbial swarmbots in microfluidic devices

Before loading swarmbot microcapsules into the culturing chamber, the microfluidic device was vacuumed for 15 min. After loading into culturing chamber, the swarmbot capsule was washed with 100 μl of M9 supplemented with different concentrations of antibiotics. Unless indicated otherwise, a programmable syringe pump (Harvard Apparatus, Pump 11 Elite) was used to control the flow pattern of either static or pulsing condition. For periodic pulsing, the flow of perfusion was set as 4 μl/min for 5 min and then delayed 25 min before the next pulse. Given a reactor volume of ~20 μl for the microfluidic device, the effective dilution rate (D) is thus ~2/h. This dilution rate is greater than the growth rate of MG1655 cells (< 1/h under our

experimental conditions), which would cause washout of the culture in a typical reactor even in the absence of an antibiotic. However, washout did not occur in our experiment, likely for two reasons. First, the swarmbot serves a reservoir that contributes to replenish the culture in the chamber. Second, the geometry of the microfluidic device, as well as the tendency of cells to stick to the PDMS surface, might have reduced the impact of flow (Appendix Fig S12).

The microfluidic device was installed in the microscopy hood and cultured at 37°C. An inverted fluorescence microscope (Leica DMI6000) was used to monitor the dynamics of population growth both inside and outside the swarmbots. Fluorescence intensity of swarmbots was analyzed using ImageJ. For swarmbots with BlaM circuit, the flow-out medium from pulsing conditions was collected in 10-ml culturing tubes that were installed in the same microscopy hood. After 16 h of experiment, cell density in the tubes was examined by measuring OD600 to confirm the performance of safeguard control (Appendix Fig S8).

### Full model for the BlaM circuit

#### *BlaM model in liquid culture*
In addition to the simplified density model described in Fig 1, we developed a more detailed model to describe the BlaM circuit dynamics. We first consider the circuit dynamics in a single compartment, where the cell density corresponds to engineered bacteria growing in the liquid culture. The model comprises a set of algebraic equations and ordinary differential equations that describe the temporal dynamics of four components: cell density ($N$), nutrient concentration ($S$), carbenicillin concentration ($A$), and β-lactamase concentration ($B$).

$$G = \frac{\mu_{max}S}{K_s + S}\left(1 - \frac{N}{N_m}\right) \tag{1.1}$$

$$L = d_A \max\left\{\frac{A}{K_{lysis} + A}\left(\frac{G}{\mu_{max}}\right) - \left(\frac{G_0}{\mu_{max}}\right), 0\right\} \tag{1.2}$$

$$\frac{dN}{dt} = GN - LN \tag{1.3}$$

$$\frac{dS}{dt} = -\alpha GN + k_r LN \tag{1.4}$$

$$\frac{dA}{dt} = -\frac{v_{max}BA}{K_{amp} + A} \tag{1.5}$$

$$\frac{dB}{dt} = -d_B B + k_{bla} LN \tag{1.6}$$

where $\mu_{max}$ is the maximum growth rate, $K_s$ is the half-maximal constant for cell growth, $N_m$ is the carrying capacity, $d_A$ is the killing rate constant by antibiotics, $K_{lysis}$ is the half-maximal constant for lysis, $\alpha$ is the rate constant of nutrient consumption, $k_r$ is the rate constant of nutrient release upon lysis, $v_{max}$ is rate constant of antibiotic degradation by BlaM, $K_{apm}$ is the half-maximal constant for enzyme reaction, $d_B$ is the rate of intrinsic BlaM degradation, and $k_{bla}$ is the rate constant of BlaM synthesis and release.

We assume that the growth rate of cells, $G$, is limited by $S$, following the Monod kinetics, and the carrying capacity ($N_m$),

following the logistic equation. We assume that the lysis rate ($L$) depends on $A$, following the Michaelis–Menten kinetics, and $G$. The dependence on $G$ is based on the observation that the killing rate by β-lactams increases with bacterial growth rate (Tuomanen *et al*, 1986). However, we made a further assumption that lysis only occurs when cells grow sufficiently fast.

Non-dimensionalization of the full BlaM model leads to the following equations:

$$g = \frac{s}{1+s}(1-n) \tag{2.1}$$

$$l = \max\left\{\frac{a}{\sigma_1+a}g - g_0, 0\right\} \tag{2.2}$$

$$\frac{dn}{d\tau} = gn - \gamma_1 ln \tag{2.3}$$

$$\frac{ds}{d\tau} = -\gamma_2 gn + \beta_1\gamma_1 ln \tag{2.4}$$

$$\frac{da}{d\tau} = -\frac{ba}{1+a} \tag{2.5}$$

$$\frac{db}{d\tau} = -\gamma_3 b + \beta_2\gamma_1 ln \tag{2.6}$$

The definitions of the dimensionless variables, their physical interpretations, and their base values are listed in Appendix Table S1.

*Two-compartment model*

To describe microbial swarmbot dynamics in the microfluidic device, we expand the full BlaM model into a two-compartment model (Fig 3A). One compartment corresponds to the swarmbot, the other the culturing chamber. Without loss of generality, we consider a single swarmbot capsule in the chamber.

In each compartment, the system dynamics are described by essentially the same set of ODEs comprising the full BlaM model. In addition, we introduce transport terms to account for diffusion of molecules and escape of cells across the shell of swarmbot capsule. We further introduce an input function $D(\tau)$ to account for dilution and replenishment of each component for the pulsing-flow growth condition. $D(\tau)$ is set to 0 for the static growth condition.

The two-compartment model consists of following equations:

Inside the microbial swarmbot (index = 1)

$$g_1 = \frac{s_1}{1+s_1}(1-n_1) \tag{3.1}$$

$$l_1 = \max\left\{\frac{a_1}{\sigma_1+a_1}g_1 - g_0, 0\right\} \tag{3.2}$$

$$\frac{dn_1}{d\tau} = g_1 n_1 - \gamma_1 l_1 n_1 + f_n(n_2 - n_1) \tag{3.3}$$

$$\frac{ds_1}{d\tau} = -\gamma_2 g_1 n_1 + \beta_1\gamma_1 l_1 n_1 + f_s(s_2 - s_1) \tag{3.4}$$

$$\frac{da_1}{d\tau} = -\frac{b_1 a_1}{1+a_1} + f_a(a_2 - a_1) \tag{3.5}$$

$$\frac{db_1}{d\tau} = -\gamma_3 b_1 + \beta_2\gamma_1 l_1 n_1 + f_b(b_2 - b_1) \tag{3.6}$$

In the chamber (index = 2)

$$g_2 = \frac{s_2}{1+s_2}(1-n_2) \tag{4.1}$$

$$l_2 = \max\left\{\frac{a_2}{\sigma_1+a_2}g_2 - g_0, 0\right\} \tag{4.2}$$

$$\frac{dn_2}{d\tau} = g_2 n_2 - \gamma_1 l_2 n_2 + f_n\frac{1}{V_R}(n_1 - n_2) - D(\tau)n_2 \tag{4.3}$$

$$\frac{ds_2}{d\tau} = -\gamma_2 g_2 n_2 + \beta_1\gamma_1 l_2 n_2 + f_s\frac{1}{V_R}(s_1 - s_2) + D(\tau)I_s - D(\tau)s_2 \tag{4.4}$$

$$\frac{da_2}{d\tau} = -\frac{b_2 a_2}{1+a_2} + f_a\frac{1}{V_R}(a_1 - a_2) + D(\tau)I_a - D(\tau)a_2 \tag{4.5}$$

$$\frac{db_2}{d\tau} = -\gamma_3 b_2 + \beta_2\gamma_1 l_2 n_2 + f_b\frac{1}{V_R}(b_1 - b_2) - D(\tau)b_2 \tag{4.6}$$

Here, $I_s$ and $I_a$ indicate concentrations of nutrient and carbenicillin in the fresh medium. Furthermore, we assume that the transport rate of each component (cells or molecules) is proportional to the concentration gradient between the two compartments. $f_n$, $f_b$, $f_s$, and $f_a$ indicate transport rate constant of the corresponding variables. The volume ratio, $V_R$, is defined as $V_2/V_1$. The transport rate constant of each component is chosen depending on the size of each component. For instance, the size of a cell is about 3 orders of magnitude larger than the other components.

**Expanded View** for this article is available online.

### Acknowledgements

We thank Anand Pai for the BlaM circuit, Frances H Arnold for the p*lux*CAT plasmid (for the QS-CAT circuit), the Shared Materials and Instrumentation Facility at Duke University for assistance in fabricating our microfluidic devices, and Cheemeng Tan, Xiao Wang, and Rob Smith for helpful comments on the manuscript. This work was partially supported by the National Science Foundation (LY, CBET-0953202), the National Institutes of Health (LY, R01GM098642, R01GM110494), the Army Research Office (#W911NF-14-1-0490), and a David and Lucile Packard Fellowship (LY).

### Author contributions

SH designed and performed the experiments, designed and fabricated the microfluidic devices, interpreted the results, and wrote the manuscript. AJL designed and performed modeling and experiments, interpreted the results, and wrote the manuscript. RT and FW assisted with experiments and manuscript revision. YZ assisted with fabrication of microfluidic devices and microcapsules. KWL assisted in manuscript revision. LY conceived the research, assisted in research design, data interpretation, model construction, and wrote the manuscript. All authors approve the manuscript.

### Conflict of interest

The authors declare that they have no conflict of interest.

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
