## [Review Process File · Molecular Systems Biology]

Coupling spatial segregation with synthetic circuits to control bacterial survival

Shuqiang Huang, Anna Jisu Lee, Ryan Tsoi, Feilun Wu, Ying Zhang, Kam W. Leong and Lingchong You

Corresponding author: Lingchong You, Duke University

Review timeline:

Submission date:	09 September 2015
Editorial Decision:	23 October 2015
Revision received:	08 December 2015
Editorial Decision:	18 January 2016
Revision received:	30 January 2016
Accepted:	03 February 2016

Editor: Maria Polychronidou

Transaction Report:

1st Editorial Decision

23 October 2015

Thank you again for submitting your work to Molecular Systems Biology. I apologize for the delayed response, which was due to the late arrival of one of the reports. We have now heard back from the two referees who agreed to evaluate your manuscript. Both reviewers appreciate that the presented system seems interesting. However, they raise a series of concerns, which should be carefully addressed in a revision of the manuscript. Their recommendations are quite clear so there is no need to repeat the points listed below.

REFeree COMMENTS

Reviewer #1:

In this manuscript, You and coworkers report the design of "Microbial Swarm Bots", a hybrid material/synthetic biology system to safeguard against the release of genetically engineered bacteria into the environment. The approach combines membrane encapsulation of bacteria engineered with genetic circuits that encourage their growth at high density, but discourage their growth at low density. The idea is interesting, and fairly innovative. The group combines the idea with modeling of the key processes that affect cell growth in these conditions and experiments using a microfluidic system and *E. coli* engineered with 3 different types of density-dependent antibiotic resistance systems. Overall, I believe the work is of interest to the Molecular Systems Biology readership, and worth of publication, given several edits to the manuscript.

First, I think that Microbial Swarm Bots is a misleading name for the system reported in this paper. "Swarm Bot" implies that the bacteria autonomously swarm, or aggregate together, like robots. However, the system reported in the manuscript requires that the microbes are placed together in a synthetic membranous structure. Then, if they escape, they are killed due to an environmental antibiotic that acts on low density bacteria. This is more like density dependent survival of environmental toxins, possibly somewhat like what biofilms or other natural multicellular structures do. I believe that the difference between what is reported as a Swarm Bot is significant, because if they did truly swarm, the system would have additional implications for safety/limiting release that might extend the results achieved in this paper. Thus, I believe the authors should choose a different name.

The models developed in the paper provide a useful guide to interpreting system performance and dependence on key parameters, especially in Figures 4 and 5. However, there is no firm or quantitative relationship between model predictions and experimental measurements. A major goal of synthetic biology, especially from the perspective of the systems biology readership of MSB, is to quantitatively predict the performance of engineered systems from the measured performance of their component parts. Thus, using the model to make quantitative predictions and examining those predictions experimentally would strengthen this manuscript significantly.

It would strengthen the paper if the authors expanded the experiment in Figure S7 and did a VR versus final density curve, to compare it to the model in Figure 1D.

On Page 9, I believe the callout to Figure 2G-H is in error.

The axes in Figures 2G-I should be labeled and the text describing this figure on main text Page 6 should discuss the values of N_{CT} observed for each circuit. Could these values be at all predicted based on independent measurements of the performance of the components?

I believe the term 'self-addiction' is improperly used in the paper. Addiction systems refer to genetically encoded self-selecting systems - i.e. those that do not require an environmental selective pressure to maintain the DNA in the population. Addiction is commonly used to refer to Toxin/Antitoxin systems for this reason. What the authors report here is more similar to a traditional selective marker. Importantly, if the antibiotic was removed from the environment, the circuits designed in this paper could readily evolve away/be lost. On the contrary, a true addiction system would be maintained in the absence of an extracellular selective agent. A true addiction system would be very interesting to use, because it might be able to restrict dispersion of engineered cells into the environment over long timescales. Thus, I think the terminology should be changed to avoid confusion.

Reviewer #2:

Summary

The authors describe three gene circuits that cause the cells to have a threshold cell density in order to proliferate. The systems are characterized in culture and then implemented in a permeable-microcapsule/microfluidics setting. Accompanying models guide experiments and provide insights into the behavior. The systems are offered as proof of principle experiments for the design of safeguarded cells that cannot proliferate upon escape from a desired environment.

General Remarks

I think this work is very creative and interesting. Engineering cells that cannot proliferate when they escape from a certain environment is an important application that synthetic biology can address. The combination of modeling and experiment is a strength of the paper. I think the paper can be strengthened in several respects but particularly by (1) quantitative indications of the robustness of the phenomena and (2) a more complete discussion/demonstration of the observed behavior of the QS-CAT and QS-BlaM systems (i.e. Figure 6) that justifies the authors' contention that the systems are working as designed. This paper will be of particular interest to the synthetic biology community.

Major Comments

1.p. 7, top of the page. The authors think that the higher survival threshold of the QS-BlaM circuit is "likely due to the two layers of density control." There are at least three other possible explanations that are simpler. First, the BlaM gene is under different promoters in the two systems. Potentially, the amount of BlaM produced in the QS-BlaM circuit is less than that produced in the BlaM circuit. Second, perhaps the export of BlaM via the non-native HlyB/D transport system is very inefficient. Third, if the Hly export tag is not cleaved upon export (I don't know if this export system does this) then the C-terminal tag could affect the catalytic activity or stability of the enzyme. Have these possibilities been examined and controlled for?

2. Figure 3E-F and Figure 5. Are these just traces of single experiments or averages of multiple experiments? If they are just single, I think multiple replications should be performed and reported. If they do represent multiple experiments, then some sort of indication of the variation (error bars or supplemental figure showing all traces) is necessary to understand the robustness of the phenomena. Also, why aren't similar graphs shown for the experiments of Figure 6?

3. Fig 6 row 3. Do the authors have a reason for the concentrated growth away from the large microcapsule? Why isn't the growth outside the microcapsule diffuse like in the other circuits. It appears that there is (perhaps) microcapsule debris so that the growth observed outside the large microcapsule is just in the small bits of broken up capsules. Can you really tell the circuit is working as expected if there isn't diffuse growth outside the capsule in the absence of Cm?

4.p. 11 lines 4-7. The authors state that QS-BlaM circuit provided safeguard control even under static conditions (Figure 6 last two rows of images). But this seems to be only weakly dependent on the presence on Cb. Do the authors have an explanation? Also, it seems like the background level of green outside the chamber in the series with no antibiotic is higher than that with 100 ug/ml Cb, which exaggerates the level of growth outside the microcapsule (but inside the chamber) in the no Cb relative to the 100ug/ml Cb. Are the authors sure these images were taken using the same settings and have had the same image manipulation performed?

Minor Comments

1. The authors remark a couple of times that the antibiotic resistance gene is not an essential feature of the system and that other density-dependent systems could be realized. I agree. But I think readers would benefit from the authors briefly describing a hypothetical example (e.g. expression of an essential enzyme under a cell density-dependent promoter), perhaps in the middle of the last paragraph of the discussion.

2. I wonder if there might be a better term than "self-addiction," but I don't have any suggestions. It took me a bit to understand what was meant by the term.

3. Figure 2G-I, the y-axes are not labeled and the x-axes are not labeled and have no numbers. I assume the graphs are OD vs. time. Are the numbers on the curves concentrations of antibiotics in ug/ml? This should be indicated.

4.p. 6 line 7. I suggest a reference for the HlyB/D transporters as this system not as widely known transport machinery in E. coli. An explanation of why the normal signal sequence of Bla was not used in this circuit would be helpful to the reader (i.e. that the HlyB/D transporter transports the protein extracellularly, unlike the natural Bla signal sequence). Also, I think it is confusing to give the gene the same name (BlaM) as that used in the BlaM circuit. I presume the gene is different because a C-terminal Hly signal peptide has been added to the end. I recommend changing the name of the gene (and the circuit) accordingly so that no one mistakenly thinks the exact same gene is used in both.

5.p. 6 middle paragraph. I agree that we would expect the three circuits to have different survival thresholds. But I think some of the reason for the BlaM circuit is not fully articulated. For the regulation for the BlaM circuit, I think it would be better to also state "enough cells have to be present to produce sufficient quantities of BlaM" (like it is stated in the second regulation layer of the QS-BlaM circuit). This more clearly articulates a similarity between the BlaM and QS-BlaM circuits (though of course there are differences).

6. There should be a reference to Figure 6 in the first full sentence at the top of page 7 (when talking about the QS-CAT bacteria).

1st Revision - authors' response

08 December 2015

Point-by-point responses to reviewers' comments

Reviewer# 1:

In this manuscript, You and coworkers report the design of "Microbial Swarm Bots", a hybrid material/synthetic biology system to safeguard against the release of genetically engineered bacteria into the environment. The approach combines membrane encapsulation of bacteria engineered with genetic circuits that encourage their growth at high density, but discourage their growth at low density. The idea is interesting, and fairly innovative. The group combines the idea with modeling of the key processes that affect cell growth in these conditions and experiments using a microfluidic system and E. coli engineered with 3 different types of density-dependent antibiotic resistance systems. Overall, I believe the work is of interest to the Molecular Systems Biology readership, and worth of publication, given several edits to the manuscript.

We appreciate the reviewer's appreciation on the novelty and significance of our work, as well as insightful comments.

First, I think that Microbial Swarm Bots is a misleading name for the system reported in this paper. "Swarm Bot" implies that the bacteria autonomously swarm, or aggregate together, like robots. However, the system reported in the manuscript requires that the microbes are placed together in a synthetic membranous structure. Then, if they escape, they are killed due to an environmental antibiotic that acts on low density bacteria. This is more like density dependent survival of environmental toxins, possibly somewhat like what biofilms or other natural multicellular structures do. I believe that the difference between what is reported as a Swarm Bot is significant, because if they did truly swarm, the system would have additional implications for safety/limiting release that might extend the results achieved in this paper. Thus, I believe the authors should choose a different name.

We understand the reservation that the reviewer has on this term, "microbial swarmbot".

The term *swarm* is typically used to describe diverse groups of entities, spanning from microbes to robots. Sometimes, it implies a critical role of motion – swarming bacteria or swarming birds. This interpretation is perhaps most consistent with the reviewer's interpretation of *swarm*. More generally, the term *swarm* refers to *collective, decentralized decision* by a group of identical or similar entities (https://en.wikipedia.org/w/index.php?title=Swarm_behaviour&redirect=no; https://en.wikipedia.org/wiki/Swarm_intelligence). An example is a group of robots interacting with each other. With this interpretation, the direct and indirect communication between the individual entities is the more critical defining feature of the collective than the physical movement.

As noted by the reviewer, in our system, the gene circuits are used to program a **collective behavior** – collective survival by a population of bacteria when present at sufficiently high densities. When integrated with spatial segregation, this property enables programming of safeguard control on the bacterial population. The diverse ways to implement this property also facilitates the modularity and generality of our technology, as illustrated by our use of multiple collective-survival circuits. This property of system is consistent with the more general interpretation of the term *swarm*. The term *swarmbot* further implies the notion of treating cells as programmable entities, which is a well-accepted notion in synthetic biology.

For these reasons, we believe our use of the term *swarmbot* is appropriate. In light of the reviewer's comments, we have clarified the rationale of this definition in the main text, which we hope would alleviate the reviewer's concern (Page 4). If the reviewer would kindly agree, we wish to keep the term *microbial swarmbot*.

The models developed in the paper provide a useful guide to interpreting system performance and dependence on key parameters, especially in Figures 4 and 5. However, there is no firm or quantitative relationship between model predictions and experimental measurements. A major goal of synthetic biology, especially from the perspective of the systems biology readership of MSB, is to quantitatively predict the performance of engineered systems from the measured performance of their component parts. Thus, using the model to make quantitative predictions and examining those predictions experimentally would strengthen this manuscript significantly.

It would strengthen the paper if the authors expanded the experiment in Figure S7 and did a VR versus final density curve, to compare it to the model in Figure 1D.

We greatly appreciate the insightful comments about simulation and experimental measurements. According to the suggestions from the reviewer, we have expanded the previous Fig S7 by providing data from additional volume ratios (V_R). The trend exhibited by results is consistent with the model prediction in Fig 1D (Appendix Fig S7).

On Page 9, I believe the callout to Figure 2G-H is in error.

The placement for the callout to the designated figures was indeed unnecessary in the main text. We removed the callout accordingly.

The axes in Figures 2G-I should be labeled and the text describing this figure on main text Page 6 should discuss the values of N_{CT} observed for each circuit.

We thank the reviewer for the suggestion. We have revised the figures accordingly and provided additional explanations on the N_{CT} values in the main text in our revised manuscript (Page 7).

Could these values be at all predicted based on independent measurements of the performance of the components?

Indeed, we believe that N_{CT} can be estimated based on the kinetic properties of individual components (e.g., synthesis rate of BlaM, enzymatic degradation rate of antibiotics, etc.). For example, we have shown that initial antibiotic concentration modulates N_{CT} with both modeling and experiment. However, predicting the N_{CT} value for specific circuits is not the objective of our study. Instead, we adopted these circuits as each is expected to exhibit collective survival. This property is the key for demonstrating the overall function of our integrated system.

I believe the term 'self-addiction' is improperly used in the paper. Addiction systems refer to genetically encoded self-selecting systems - i.e. those that do not require an environmental selective pressure to maintain the DNA in the population. Addiction is commonly used to refer to Toxin/Antitoxin systems for this reason. What the authors report here is more similar to a traditional selective marker. Importantly, if the antibiotic was removed from the environment, the circuits designed in this paper could readily evolve away/be lost. On the contrary, a true addiction system would be maintained in the absence of an extracellular selective agent. A true addiction system would be very interesting to use, because it might be able to restrict dispersion of engineered cells into the environment over long timescales. Thus, I think the terminology should be changed to avoid confusion.

We agree that other words can more accurately describe the behavior of our system. In particular, we used “self-addition” and “density-dependent survival” interchangeably in our manuscript, which has caused confusion. In light of the suggestion, we have replaced both terms with “collective survival” to be more precise and coherent.

Reviewer #2:

Summary

The authors describe three gene circuits that cause the cells to have a threshold cell density in order to proliferate. The systems are characterized in culture and then implemented in a permeable-microcapsule/microfluidics setting. Accompanying models guide experiments and provide insights

into the behavior. The systems are offered as proof of principle experiments for the design of safeguarded cells that cannot proliferate upon escape from a desired environment.

We appreciate the reviewer's acknowledgement of the novelty and significance of our work. In light of the reviewer's comments, we have thoroughly revised the manuscript and provided additional data analysis as needed.

General Remarks

I think this work is very creative and interesting. Engineering cells that cannot proliferate when they escape from a certain environment is an important application that synthetic biology can address. The combination of modeling and experiment is a strength of the paper. I think the paper can be strengthened in several respects but particularly by (1) quantitative indications of the robustness of the phenomena and (2) a more complete discussion/demonstration of the observed behavior of the QS-CAT and QS-BlaM systems (i.e. Figure 6) that justifies the authors' contention that the systems are working as designed. This paper will be of particular interest to the synthetic biology community.

We greatly appreciate the reviewer's positive assessment and insightful comments. According to the suggestions, we have (1) extended the indication of safeguard performance as a function of volume ratio (V_R , Appendix Fig S7), (2) quantified the safeguard performance in Fig 6 (Appendix Fig S11), and (3) elaborated on the observed behaviors of the QS-CAT and QS-BlaM systems in the supplementary information (Appendix Fig S11).

Major Comments

1.p. 7, top of the page. The authors think that the higher survival threshold of the QS-BlaM circuit is "likely due to the two layers of density control." There are at least three other possible explanations that are simpler. First, the BlaM gene is under different promoters in the two systems. Potentially, the amount of BlaM produced in the QS-BlaM circuit is less than that produced in the BlaM circuit. Second, perhaps the export of BlaM via the non-native HlyB/D transport system is very inefficient. Third, if the Hly export tag is not cleaved upon export (I don't know if this export system does this) then the C-terminal tag could affect the catalytic activity or stability of the enzyme. Have these possibilities been examined and controlled for?

We appreciate the reviewer's insights on alternative explanations for our observation. We agree that multiple differences in the collective-survival systems could contribute to the varying survival thresholds. We have examined and incorporated the reviewer's suggestions into the main text (Page 7).

2. Figure 3E-F and Figure 5. Are these just traces of single experiments or averages of multiple experiments? If they are just single, I think multiple replications should be performed and reported. If they do represent multiple experiments, then some sort of indication of the variation (error bars or supplemental figure showing all traces) is necessary to understand the robustness of the phenomena. Also, why aren't similar graphs shown for the experiments of Figure 6?

We thank the reviewer's suggestion on justifying the data reproducibility. Fig 3 and Fig 5 each displays the data from a single experiment. We had indeed conducted replicate experiments but did not present the data to simplify presentation. We have now presented the data from these replicate experiments in Appendix Fig S9 (for Fig 3) and in Appendix Fig S10 (for Fig 5). We have also presented quantification of data from Fig 6 in Appendix Fig S11.

3. Fig 6 row 3. Do the authors have a reason for the concentrated growth away from the large microcapsule? Why isn't the growth outside the microcapsule diffuse like in the other circuits. It appears that there is (perhaps) microcapsule debris so that the growth observed outside the large microcapsule is just in the small bits of broken up capsules. Can you really tell the circuit is working as expected if there isn't diffuse growth outside the capsule in the absence of Cm?

We have re-examined our time-lapse data and conducted another replicate experiment for Figure 6 row 3 (Appendix Fig 12D). The apparent patchy growth was unlikely from the debris of the capsule, but instead due to heterogeneity of the PDMS surface. Effects of this heterogeneity were mostly evident in the absence of the antibiotic. This patchy growth was also evident for other circuits at

earlier time points, including the BlaM cells (before 16th hour). We have included additional experimental data to illustrate these points (Appendix Fig S12).

In the presence of antibiotics, the escaped cells could not survive/proliferate due to their low density (Fig 2H), despite the heterogeneity of the PDMS surface. We note that this technical issue (surface heterogeneity) is not critical for the demonstration of our designed circuit function.

4.p. 11 lines 4-7. The authors state that QS-BlaM circuit provided safeguard control even under static conditions (Figure 6 last two rows of images). But this seems to be only weakly dependent on the presence on Cb. Do the authors have an explanation? Also, it seems like the background level of green outside the chamber in the series with no antibiotic is higher than that with 100 ug/ml Cb, which exaggerates the level of growth outside the microcapsule (but inside the chamber) in the no Cb relative to the 100ug/ml Cb. Are the authors sure these images were taken using the same settings and have had the same image manipulation performed?

These images were taken using the same settings and processed using the same procedures. The apparent issue raised by the reviewer was due to the low resolution of images in the original submission. We have uploaded higher-resolution images, and analyzed the background signals (as Fig 1 below), which showed that the backgrounds were indeed quite similar.

Fig 1. Analysis of background signals for MSBs with the QS-BlaM circuit. The red arrows in the insets indicated the four positions used for the analysis, and the plot data was extracted from GFP channel alone. Error bars indicate standard deviations.

Minor Comments

1. The authors remark a couple of times that the antibiotic resistance gene is not an essential feature of the system and that other density-dependent systems could be realized. I agree. But I think readers would benefit from the authors briefly describing a hypothetical example (e.g. expression of an essential enzyme under a cell density-dependent promoter), perhaps in the middle of the last paragraph of the discussion.

We thank the reviewer for the insightful suggestions. We have discussed additional circuits that can achieve collective survival without exogenous addition of antibiotics (Page 12).

2. I wonder if there might be a better term than "self-addiction," but I don't have any suggestions. It took a bit to understand what was meant by the term.

We agree. We have replaced the term "self-addiction" with "collective survival".

3. Figure 2G-I, the y-axes are not labeled and the x-axes are not labeled and have no numbers. I assume the graphs are OD vs. time. Are the numbers on the curves concentrations of antibiotics in ug/ml? This should be indicated.

Thanks and we have corrected the text accordingly.

4.p. 6 line 7. I suggest a reference for the HlyB/D transporters as this system not as widely known transport machinery in E. coli. An explanation of why the normal signal sequence of Bla was not used in this circuit would be helpful to the reader (i.e. that the HlyB/D transporter transports the protein extracellularly, unlike the natural Bla signal sequence). Also, I think it is confusing to give

the gene the same name (BlaM) as that used in the BlaM circuit. I presume the gene is different because a C-terminal Hly signal peptide has been added to the end. I recommend changing the name of the gene (and the circuit) accordingly so that no one mistakenly thinks the exact same gene is used in both.

We appreciate the insightful comments from the reviewer. We have included a reference for the HlyB/D transporter system in the main text (Chervaux et al, 1995), as well as more detailed explanations of its function. The BlaM in the BlaM circuit and that in the QS-BlaM are indeed slightly different. The BlaM in the QS-BlaM circuit is fused to a HlyAs secretion tag, which allows its transport through HlyB/D apparatus (Pai et al, 2012; Tanouchi et al, 2012). However, for simplicity, we wish to keep the current name for the system but with further clarifications (Page 6 and Appendix Fig S1). We hope the clarification in our revised manuscript will remove any potential confusion about the system implementation.

5.p. 6 middle paragraph. I agree that we would expect the three circuits to have different survival thresholds. But I think some of the reason for the BlaM circuit is not fully articulated. For the regulation for the BlaM circuit, I think it would be better to also state "enough cells have to be present to produce sufficient quantities of BlaM" (like it is stated in the second regulation layer of the QS-BlaM circuit). This more clearly articulates a similarity between the BlaM and QS-BlaM circuits (though of course there are differences).

We agree with the reviewer and have incorporated these suggestions.

6. There should be a reference to Figure 6 in the first full sentence at the top of page 7 (when talking about the QS-CAT bacteria).

After reviewing this section, we believe that the reviewer was asking us to refer to Figure 2 on Page 7 because we did not start to discuss the results of Figure 6 on that page yet. We have revised the manuscript accordingly.

Reference

Chervaux C, Sauvonnet N, Le Clainche A, Kenny B, Hung AL, Broome-Smith JK, Holland IB (1995) Secretion of active beta-lactamase to the medium mediated by the Escherichia coli haemolysin transport pathway. *Molecular & general genetics : MGG* **249**: 237-245

Pai A, Tanouchi Y, You LC (2012) Optimality and robustness in quorum sensing (QS)-mediated regulation of a costly public good enzyme. *P Natl Acad Sci USA* **109**: 19810-19815

Tanouchi Y, Pai A, Buchler NE, You LC (2012) Programming stress-induced altruistic death in engineered bacteria. *Mol Syst Biol* **8**

Thank you again for submitting your work to Molecular Systems Biology. We have now heard back from the two referees who were asked to evaluate your manuscript. As you will see below, Reviewer #2 still expresses a remaining concern, which we would ask you to address in a revision of the work.

In particular, reviewer #2 mentions that further controls excluding alternative explanations for the higher survival threshold of the QS-BlaM circuit need to be included. We have circulated the reports to both reviewers as part of our 'pre-decision cross-commenting' policy. During this process, Reviewer #1 agreed with Reviewer #2 that these control experiments should be performed.

 REFEE COMMENTS

Reviewer #1:

The authors has satisfactorily responded to the referee comments and I support publication in MSB.

Reviewer #2:

The authors have satisfactorily addressed all my concerns except for the most important one (my major point #1; see below).

As a result I am rating "Validity of conclusions drawn" and Suitability of Publication" as "Low" solely for the lack of proper controls on this important experiment. The paper is otherwise very good.

1) In my major point #1, I pointed out that there are three possible alternative explanations as to why the QS-BlaM circuit had a higher survival threshold. The authors now include these three possible mechanisms as possible contributors to the observed behavior, but still conclude that "a major factor contributing to the higher NCT was likely the two layers of density control of the QS-BlaM circuit."

In my view, it is not sufficient to merely acknowledge the alternative explanations. They need to be controlled for. The authors are trying to prove that their circuit works as designed. Any of the three mechanisms (which are simpler explanations) is sufficient to explain the data by themselves. I feel that if the authors want to claim that their mechanism is contributing to the observed behavior as designed, these alternative mechanisms need to be controlled for. Since the authors' claims are central to the paper, they data need to be substantiated by the appropriate controls.

2) Minor point: Figure S11. The legend or figure itself should indicate that the dashed lines are the chamber and the solid lines are MSB (I assume this is the case)

 2nd Revision - authors' response

30 January 2016

Detailed response to reviewer 2's comments

Reviewer #2:

The authors have satisfactorily addressed all my concerns except for the most important one (my major point #1; see below).

As a result I am rating "Validity of conclusions drawn" and Suitability of Publication" as "Low" solely for the lack of proper controls on this important experiment. The paper is otherwise very good.

1) In my major point #1, I pointed out that there are three possible alternative explanations as to why the QS-BlaM circuit had a higher survival threshold. The authors now include these three possible mechanisms as possible contributors to the observed behavior, but still conclude that "a major factor contributing to the higher NCT was likely the two layers of density control of the QS-BlaM circuit."

In my view, it is not sufficient to merely acknowledge the alternative explanations. They need to be controlled for. The authors are trying to prove that their circuit works as designed. Any of the three mechanisms (which are simpler explanations) is sufficient to explain the data by themselves. I feel that if the authors want to claim that their mechanism is contributing to the observed behavior as designed, these alternative mechanisms need to be controlled for. Since the authors' claims are central to the paper, they data need to be substantiated by the appropriate controls.

We thank the reviewer again for his/her time in evaluating the work in depth. We appreciate the reviewer's insightful comment. In light of this comment, we wish to clarify the rationale of using the QS-BlaM circuit and relevant data interpretation.

As detailed in the manuscript, the central design concept is the integration of two aspects: (1) a circuit to program collective survival in a bacterial population, and (2) spatial segregation by encapsulation. We used three circuits (including QS-BlaM) that demonstrate the generality of the design concept. In this conceptual framework, the most critical aspect of a circuit is the ability to program collective survival. Our data in Figure 2 indeed demonstrated this function for all circuits, despite the difference between their regulation.

We should clarify that the detailed characterization of the circuit (including characterization of circuit components) has been thoroughly established in our previous work (Pai et al, 2012). That work uses the QS-BlaM circuit to explore evolutionary advantage of QS in regulating a public good.

We completely agree with the reviewer that alternative mechanisms could contribute to the circuit function. Our data cannot exclude these contributing factors completely, nor do we claim so. However, we now see the need to provide direct evidence that QS regulates the survival threshold. To this end, we have conducted additional experiments to further demonstrate the functionality of QS-BlaM circuit. The circuit is designed such that QS is fully activated when both LuxI (by IPTG/arabinose) and LuxR (by aTc) are induced. As shown in Appendix Fig S13, induction of QS reduced the survival threshold by 2.2 fold in comparison to the control where QS was not induced. We note that, in the absence of inducers, the basal level activation of QS was able to drive leaky expression of BlaMs, which also enabled collective survival (with a high threshold).

Finally, in light of reviewer's comment, we realized that we unnecessarily emphasized the notion of two layers of control in collective survival in the previous submissions. To this end, we have revised the text to remove this point, which is not central to our conclusions.

2) Minor point: Figure S11. The legend or figure itself should indicate that the dashed lines are the chamber and the solid lines are MSB (I assume this is the case)

We thank the reviewer to point this out. We have corrected the text to clarify the difference between the solid and dashed lines.

Again, we greatly appreciate the time and efforts of you and referees for evaluating our work again. We hope you and the reviewers will find the revised manuscript ready for publication in *Molecular Systems Biology*.